# A Similarity Measure-Based Approach Using RS-fMRI Data for Autism Spectrum Disorder Diagnosis

**DOI:** 10.3390/diagnostics13020218

**Published:** 2023-01-06

**Authors:** Xiangfei Zhang, Shayel Parvez Shams, Hang Yu, Zhengxia Wang, Qingchen Zhang

**Affiliations:** 1School of Cyberspace Security, Hainan University, Haikou 570228, China; 2School of Computer Science and Technology, Shandong Technology and Business University, Yantai 264005, China; 3School of Computer Science and Technology, Hainan University, Haikou 570228, China

**Keywords:** autism spectrum disorder, few-shot learning, computer-aided diagnostics, RS-fMRI

## Abstract

Autism spectrum disorder (ASD) is a lifelong neurological disease, which seriously reduces the patients’ life quality. Generally, an early diagnosis is beneficial to improve ASD children’s life quality. Current methods based on samples from multiple sites for ASD diagnosis perform poorly in generalization due to the heterogeneity of the data from multiple sites. To address this problem, this paper presents a similarity measure-based approach for ASD diagnosis. Specifically, the few-shot learning strategy is used to measure potential similarities in the RS-fMRI data distributions, and, furthermore, a similarity function for samples from multiple sites is trained to enhance the generalization. On the ABIDE database, the presented approach is compared to some representative methods, such as SVM and random forest, in terms of accuracy, precision, and F1 score. The experimental results show that the experimental indicators of the proposed method are better than those of the comparison methods to varying degrees. For example, the accuracy on the TRINITY site is more than 5% higher than that of the comparison method, which clearly proves that the presented approach achieves a better generalization performance than the compared methods.

## 1. Introduction

Children with autism spectrum disorder (ASD) present a range of phenotypes, such as social and communication difficulties, restricted interests, and thinking function loss [1]. Typically, autistic children have lifelong mental retardation, and their life quality is often quite low. More unfortunately, ASD cannot be cured so far. Nevertheless, diagnosis and prompt intervention as early as possible play an important role in improving ASD children’s life quality. Neuroimaging has significantly helped us to understand the underlying pathological mechanisms of brain diseases [2,3,4,5], and, therefore, it also has been employed to diagnose ASD [6,7,8].

In neuroimaging, resting-state functional magnetic resonance imaging (RS-fMRI) utilizes blood oxygen level-dependent (BOLD) signals to explore the biomarkers of nervous system diseases [9,10,11,12]. Recently, the RS-fMRI-based ASD diagnosis has made significant progress [13,14,15,16]. For instance, Zhao et al. [14] presented a multi-view high-order functional connectivity network (FCN) based on the RS-fMRI data for ASD vs. the normal control (NC) classification. These approaches were designed based on single-site data, and, therefore, they cannot be generalized to other sites’ data because of the heterogeneity of the data from different sites. Additionally, the approaches developed from a small number of samples may present overfitting.

The RS-fMRI samples from multiple sites are somewhat heterogeneous due to the scanner type and imaging acquisition protocol. In particular, heterogeneity of the RS-fMRI samples tends to cause low generalization performance [17,18,19]. To deal with the problem caused by heterogeneity, many diagnostic modelings of ASD depending on samples from multiple sites have been explored [20,21]; these can be roughly categorized into two types. The first type ignores the heterogeneity [19,20] by assuming that samples from multiple sites are collected from the same or similar distribution. For instance, Brown et al. [21] proposed the element-wise layer for DNNs to predict ASD without considering the heterogeneity of data from different sites. The other type aims to avoid the adverse effect of data heterogeneity on the results [22,23,24]. For example, Niu et al. [22] proposed a multi-channel deep attention neural network to capture the correlations in multi-site data by integrating multi-layer neural networks, attention mechanisms, and feature fusion. However, these methods need extensive data that are hard to collect.

To improve generalization on the limited number of RS-fMRI samples, this study presents a similarity measure-based method for early ASD diagnosis. First, a Siamese network is devised to reduce the negative effect of heterogeneity on the performance of the model. Afterward, we design an independent objective function for each training site, and the total objective function is summed to train the parameters based on the back-propagation algorithm. Finally, a few samples, which the model has never seen before, are used to fine-tune the parameters with the aim of making the model adapt to new samples.

## 2. Problem Formulation

To improve the generalization and accuracy of ASD diagnoses, we follow the few-shot learning idea [25,26] to design the similarity measure-based approach in this study. Specifically, few-shot learning is formulated as learning a classifier to recognize the remaining samples given a few labeled samples for each class in the target site. If the classifier is directly trained utilizing traditional optimization algorithms, it is hard to obtain satisfactory performance due to the heterogeneity. Under the few-shot learning strategy, some imaging sites are used to train the parameters, while others are used to fine-tune the model parameters and evaluate the performance of the model. Accordingly, the whole dataset used in this study is partitioned into training sites, target sites, and a baseline site (set). In particular, the training sets are split further into meta-training sets and meta-test sets, and we aim to match the samples of the target site to the baseline site accurately.

## 3. Materials and Methods

This section illustrates the proposed method in detail, especially the Siamese network and the training strategy.

### 3.1. Materials

In this work, we sampled the studied RS-fMRI data from the Autism Brain Imaging Data Exchange (ABIDE), which aims to facilitate discovery and comparison in ASD research. We included data from the C-PAC preprocessing pipeline of ABIDE [27]. The data were excluded when the scanning data was missing from the original imaging data at some time points. After preprocessing, the RS-fMRI data from 12 imaging sites (402 ASD patients and 423 NCs) were included in the final analysis. The detailed demographic information, including age and sex, is summarized in Table 1. In this work, the AAL atlas [28] with 116 brain regions is used for brain parcellation.

### 3.2. Constructing Dynamic FCN

Functional connectivity exploits the temporal correlation of BOLD signals in different brain regions to demonstrate how structural segregation and functionally specialized brain regions interact. FCNs are of great significance for discovering the functional organization of the human brain and for finding biomarkers for neuropsychiatric diseases [10,29,30]. Studies have shown that dynamic functional connectivity patterns have significant contributions to the diagnosis of neurological diseases [9,31,32].

The sliding window strategy is a popular method for constructing a dynamic FCN (D-FCN). The illustration of constructing the D-FCN is shown in Figure 1. A D-FCN is a sequential collection of sub-networks created by dividing the entire RS-fMRI time series into multiple overlapping sub-segments, each of which is constructed as a sub-network that reflects short-term correlations. The short-term correlation in *k*-th window is calculated by Pearson’s correlation coefficient, as follows:(1)$$FC_{ij}\left(k\right)=\frac{\sum_{m=1}^{M}\left(x_{im}\left(k\right)-{\bar{x}}_{i}\left(k\right)\right)\left(x_{jm}\left(k\right)-{\bar{x}}_{j}\left(k\right)\right)}{\sqrt{\sum_{m=1}^{M}{\left(x_{im}\left(k\right)-{\bar{x}}_{i}\left(k\right)\right)}^{2}}\sqrt{\sum_{m=1}^{M}{\left(x_{jm}\left(k\right)-{\bar{x}}_{j}\left(k\right)\right)}^{2}}}$$
where *M* is the length of a segment, $x_{i}\left(k\right)$ denotes the BOLD signals of the *i*-th ROI, ${\bar{x}}_{i}\left(k\right)$ is the average value of all elements in $x_{i}\left(k\right)$, and $x_{im}\left(k\right)$ is the *m*-th element in $x_{i}\left(k\right)$. Thus, a sub-FCN of a D-FCN is constructed as $D\left(k\right)={\left[FC_{ij}\left(k\right)\right]}_{1⩽k⩽K}$, and the corresponding D-FCN can be represented as D=[D(1),…,D(k),…,D(K)], where *K* is the total number of segments.

In this work, the mean matrix of the D-FCN [14,15,33] is used as the input. Taking into account the symmetry of the functional connectivity matrix, the off-diagonal triangular parts on it are vectored as the feature vector.

### 3.3. Siamese Network Framework

Recently, deep neural networks have been used broadly in various fields [34,35,36,37,38]. However, training neural networks requires abundant data. For some fields, it is difficult to collect a large amount of data. Therefore, few-shot learning [25,26] is proposed to solve this problem, and reliable results have been obtained in multitudinous studies. Few-shot learning is usually implemented based on metric learning methods, such as Prototype networks and Siamese networks.

As shown in Figure 2, the Siamese network [39,40] consists of two sub-networks that share parameters. In this work, the Siamese network consists of two identical feature extractors, an autoencoder, and fully connected layers; the input is a 6670-dimensional feature vector extracted from each subject, and the output is the similarity of two subjects. The calculation similarity unit is formulated as:(2)$$dist\left({\hat{x}}_{1},{\hat{x}}_{2}\right)=\left|\right|{\hat{x}}_{1}-{\hat{x}}_{2}{\left|\right|}_{1}$$
where ${\hat{x}}_{1}$ and ${\hat{x}}_{2}$ are the outputs of the two feature extractors, respectively. In order to intuitively compare, the sigmoid function is used to map the $dist\left({\hat{x}}_{1},{\hat{x}}_{2}\right)$ within the range (0, 1).

The loss function is important for artificial neural networks to generate separable representations for unseen classes. In this work, the mean square error is employed as the loss function:(3)$$L=\frac{1}{2}{\left(dist({\hat{x}}_{1},{\hat{x}}_{2})-y\right)}^{2}$$
where *y* is the label, indicating whether the two subjects are from the same category, that is:$$y=\left\{\begin{array}{c} 0,\;\mathrm{if}\;\mathrm{the}\;\mathrm{two}\;\mathrm{subjects}\;\mathrm{are}\;\mathrm{from}\;\mathrm{the}\;\mathrm{same}\;\mathrm{category}\;\\ 1,\;\mathrm{otherwise}\; \end{array}\right.$$

As illustrated in Figure 2, the training Siamese network needs to input paired samples and optimize the model parameters by minimizing the loss function. In the settings of this section, the similarity is measured by the distance (Equation (Equation 2)), so when minimizing the loss function (Equation (Equation 3)), the distance between paired samples from the same category should be small, and the distance between paired samples from different categories should be large.

In the test phase, the test dataset is divided into a support set and a query set. Suppose that there are *C* categories in the data set and *N* samples in the support set of each category. This few-shot learning classification task is called a *C*-way *N*-shot task. Figure 3 shows the test flowchart of the trained Siamese network model. In our case, we have two categories (NC and ASD), so it is a two-way *N*-shot task. Given a query sample *Q* from the query set, each support set has *N* support samples. The distances/similarities between the query sample *Q* and the samples in the two support sets are calculated to predict the label of sample *Q*.

### 3.4. Few-Shot Training Strategy

Because the underlying pathology of the ASD cross-sites is similar, it can be reasonably assumed that the data extracted from multiple sites share an inherent underlying data structure [23]. The baseline set is used in both the training and final performance test. In other words, we use the data of a specific site as the baseline and compare the subjects from other imaging sites with the baseline. Our goal is to train the model parameters so that the distance/similarity comparison between other site data and baseline site data can achieve satisfactory results. That is to say, in addition to training the model parameters, a site should be found as the baseline site during the training of the model, which will be used as the support set in the test phase.

Unlike the traditional few-shot task, called the *C*-way *N*-shot task, a specific site is selected as the baseline set. It is worth noting that the number of NC samples in the baseline set may not be equal to the number of ASD samples. Therefore, it can be called a *C*-way (*C* = 2 in our case) task, but the *N* is uncertain. In this work, we employ the prototype network [41] for reference to convert the few-shot task into a two-way one-shot task. Specifically, the average abstract features (extracted by feature extractor) of the NC/ASD samples in the baseline set as the prototypical features are used to calculate the similarity with other samples.

During training, the data for each site for training is partitioned into a meta-training set and meta-test set. The training process is shown in Figure 4. The model parameters are determined by minimizing the loss between the meta-training set and the baseline set; the meta-test set is used to verify the performance of the model in each iteration. Each training site will obtain an independent loss. Suppose that there are *K* sites for training, the sum of *K* losses is used as the total loss for the tuning parameters:(4)$$L_{\mathrm{total}\;}=\sum_{k=1}^{K}L_{k}$$

Using the total loss as the objective function can prevent the model from overfitting the data of a certain site during the training process and promote the model’s learning of the “basic concept” of the similarity.

### 3.5. Experimental Setup

We selected five imaging sites as the baseline set and training sets, UCLA, UM, USM, NYU, and YALE. In each training set, 70% of the samples are used to train the model parameters, and the remaining 30% are used to evaluate the performance of the model during training. One of the five sites is selected as the baseline set, and the other four sites are selected to be the training sets during the training process. Determine the model parameters by minimizing the loss between the baseline set and the training sets.

Additionally, another seven imaging sites are used to test the performance of the final model, which are independent of the training sets. The test strategy is to predict the category by comparing the similarity between the target subject and the NC and ASD samples in the baseline set. Specifically, if the similarity between the target subject and the NC samples in the baseline set is higher, the label is predicted to be NC; otherwise, the label is ASD.

In this article, the accuracy, precision, and F1 score are used as the criteria for experimental performance evaluation.

## 4. Results

### 4.1. Classification Performance on Meta-Test Sets

When one site was selected as the baseline site, the average meta-test performance of the other four sites is presented in Table 2, and, in turn, each site except for the target sites was selected as the baseline site.

From Table 2, we can make the following observations. First, the selected baseline site affects the performance of the model. Due to the heterogeneity of the data among different imaging sites, it is difficult for the learner to make the loss function of each site converge to a certain range, although the total loss may converge. Therefore, when different baseline sets are selected, the learner will update the model parameters through different loss values. As a result, changing the baseline set will change the classification performance of the model. Second, the proposed model is effective on the training sites. Although the meta-test sets are selected from the training sets, they are not used to update and tune the model parameters. Each accuracy is greater than 60% in terms of classification accuracy, indicating that the model is effective on training sites. Finally, from the results, the average accuracy is between 60% and 70%, which can reasonably be considered to mean that the model does not overfit or underfit the data in training sites.

The loss curves are shown in Figure 5. The training loss is based on the following experimental settings. The RS-fMRI data from the YALE imaging site are selected as the baseline set; the data from the UCLA, UM, USM, and NYU imaging sites are selected as training sets. As can be seen from Figure 5, the loss function converges well.

### 4.2. Generalization Performance on Target Sites

To validate the generalization performance of the proposed model, we compared the model with several classical methods including the support vector machine (SVM), stacked autoencoder (SAE), and random forest (RF). The SVM and RF are implemented using the Python-based sklearn library, and default parameters are used. The SAE consists of three full connection layers with tanh activation function and uses two full connection layers for classification.

In the current experiment, the RS-fMRI data from the YALE imaging site are selected as the baseline set for the proposed method, and several subjects’ data from each target site are selected to fine-tune the model parameters. In order to avoid the problem of a sample imbalance, some data are randomly removed from the categories with plenty of samples to make equal the number of ASD and NC samples. For consistency comparison, the RS-fMRI data from the five imaging sites (i.e., UCLA, UM, USM, NYU, and YALE) were combined for training the SVM, RF, and SAE. The data from the target site are used as a test set to evaluate performance. In the training process, 70% of the training samples are used to train the comparison methods’ parameters, and the remaining 30% are used to evaluate the performances of the models during training. The results are summarized in Table 3. From the results, the ability of the proposed method to generalize to other imaging sites outperforms the competing methods.

## 5. Discussion

In light of their simplicity and ease of implementation, classic machine learning methods, such as the SVM and RF, have been widely used in previous RS-fMRI studies. However, to avert inter-site heterogeneity, these methods are often modeled and validated based on single-site data, which restricts their generalization to other imaging sites. Since the SAE contributes to easy overfitting when dealing with small sample size data, we merge the data from the training sites for training the SAE. In general, the SAE reshapes feature patterns in vector form to learn more informative high-level features for diagnosing ASD, but it is still difficult to generalize to other imaging sites in the face of heterogeneous data. Few-shot learning occurs by transferring knowledge learned during training tasks to unseen tasks. In our case, we chose four training sites and one baseline site, and we expected the model to learn general concepts to distinguish ASD from NC. This general capability may not be sufficient to deal with some unique characteristics of the target site, as fine-tuning of model parameters is required using a small amount of data from the query site. From the experimental results summarized in Table 3, the generalization performance of the proposed model outperforms the comparison methods, indicating that our method is effective for classifying ASD based on multi-site data.

It can be seen from Figure 5 that, after the training step is about 100, the loss value tends to be stable. This shows that the proposed method can converge rapidly. In addition, the trends of the four curves are similar, and there is no huge difference in the loss value. Except for one curve, the other three curves are almost coincidental. This shows that the model is not overfitted to the particular training station and is balanced on multiple training sites.

## 6. Conclusions

This paper presents a similarity measure-based approach for ASD diagnosis with RS-fMRI data. A unique property of this study is that samples from multiple sites are used to co-learn a similarity function with the baseline site, enabling the presented approach to achieve good generalization on unseen samples from target sites. Extensive experiments on the ABIDE I dataset show that the proposed approach has a robust generalization performance with comparable diagnostic accuracy in comparison to several well-established methods.

In a previous study, the ComBat harmonization method was used to eliminate the impact of data heterogeneity among sites [42]. In future work, we consider combining small sample learning and ComBat to better eliminate the adverse impact of data heterogeneity between sites. In addition, this work has yielded satisfactory results only using RS-fMRI data, but some recent studies have shown that it is possible to use multimodal data to diagnose neurological diseases, such as combining RS-fMRI data and structural MRI data for disease diagnosis. Using multimodal data to diagnose neurological diseases will be the focus of our future work. 

## Figures and Tables

**Figure 1 diagnostics-13-00218-f001:**
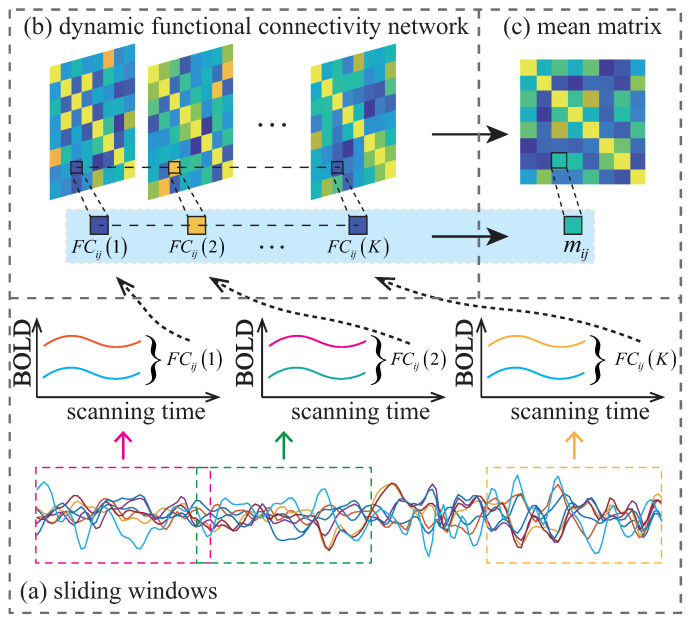
The illustration of constructing the D-FCN. The $FC_{ij}{\left(k\right)}_{1⩽k⩽K}$ denotes the temporal correlation of the *i*-th and *j*-th ROI in the *k*-th segment.

**Figure 2 diagnostics-13-00218-f002:**
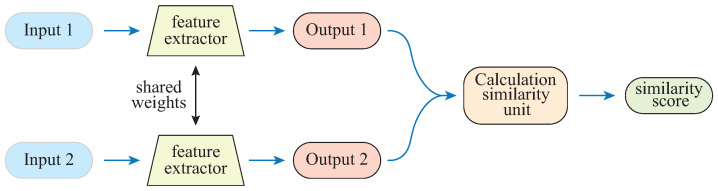
Illustration of the Siamese network.

**Figure 3 diagnostics-13-00218-f003:**
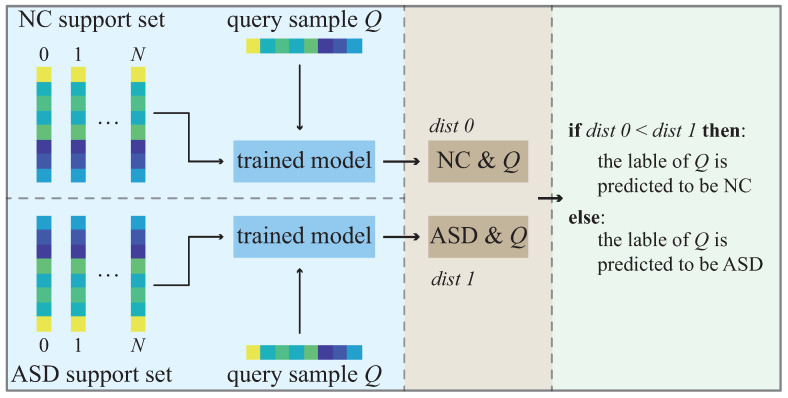
Test flowchart of the Siamese network.

**Figure 4 diagnostics-13-00218-f004:**
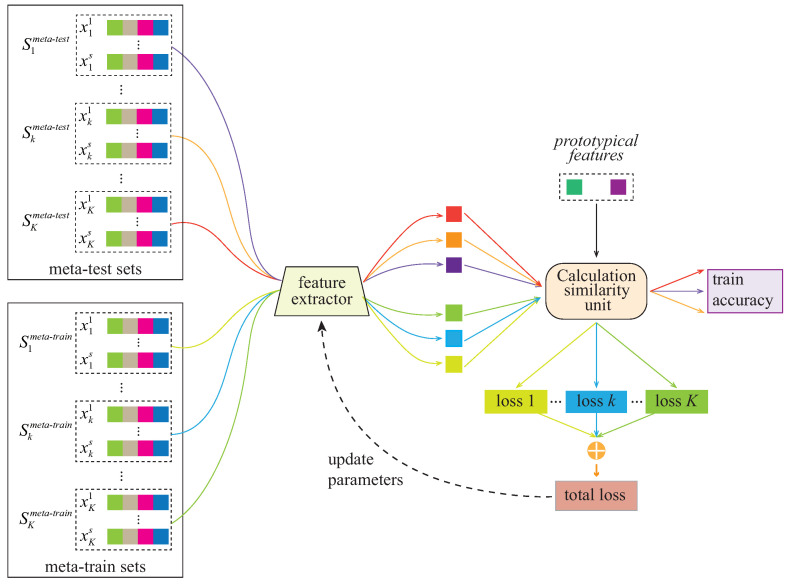
Training flowchart of the Siamese network. The *prototypical features* are extracted from the baseline site.

**Figure 5 diagnostics-13-00218-f005:**
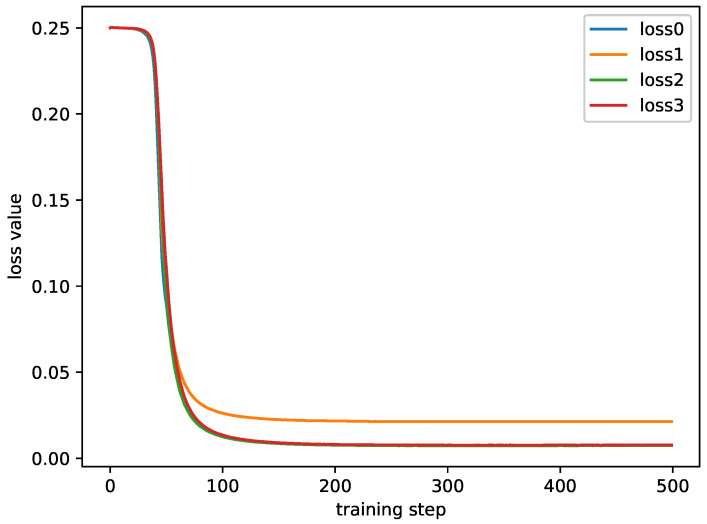
The training losses. The loss0, loss1, loss2, and loss3 are the training losses on the UCLA, UM, USM, and NYU sites, respectively.

**Table 1 diagnostics-13-00218-t001:** Demographic information of the subjects from 12 imaging sites. Average (SD) (SD, standard deviation); M/F: male/female.

	ASD		NC	
**Site**	**Age Avg (SD)**	**Count: M/F**	**Age Avg (SD)**	**Count: M/F**
KKI	10.2 (1.5)	12 / 3	10.1 (1.2)	18 / 7
OHSU	11.4 (2.2)	12 / 0	10.1 (1.1)	14 / 0
OLIN	16.5 (3.4)	16 / 3	16.7 (3.6)	13 / 2
TRINITY	16.8 (3.2)	22 / 0	17.0 (3.8)	24 / 0
MAXMUN	26.1 (14.9)	21 / 3	24.5 (9.1)	25 / 1
LEUVEN	17.8 (5.0)	26 / 3	18.3 (5.1)	28 / 5
PITT	19.8 (8.5)	16 / 3	18.0 (6.7)	14 / 5
UCLA	13.0 (2.5)	48 / 6	13.0 (1.9)	38 / 6
UM	13.0 (2.3)	55 / 9	14.9 (3.6)	56 / 17
USM	23.0 (7.4)	44 / 0	21.3 (8.4)	25 / 0
NYU	14.4 (6.9)	62 / 10	15.6 (6.1)	72 / 25
YALE	12.7 (3.0)	20 / 8	12.7 (2.8)	20 / 8

**Table 2 diagnostics-13-00218-t002:** The performance on meta-test sets.

Baseline Site	Accuracy (%)	Precision (%)	F1 Score (%)
UCLA	61.30 ± 1.61	60.87 ± 1.60	62.00 ± 3.76
UM	68.83 ± 4.75	67.35 ± 5.49	70.51 ± 3.27
USM	65.82 ± 3.84	66.41 ± 4.43	65.25 ± 4.08
NYU	65.19 ± 6.12	63.27 ± 6.73	69.00 ± 4.67
YALE	69.59 ± 1.44	72.16 ± 7.00	68.42 ± 3.51

**Table 3 diagnostics-13-00218-t003:** Performance comparisons between the proposed model and competing methods.

Site	Method	Accuracy (%)	Precision (%)	F1 Score (%)
TRINITY	RF	61.36	64.71	56.41
	SVM	68.18	70.00	66.67
	SAE	68.18	72.22	65.00
	Ours	73.68	84.21	80.00
OLIN	RF	73.33	73.33	73.33
	SVM	73.33	70.59	75.00
	SAE	73.33	73.33	73.33
	Ours	80.76	76.92	78.57
OHSU	RF	33.33	33.33	33.33
	SVM	54.17	53.33	59.26
	SAE	41.67	40.00	36.36
	Ours	57.14	57.14	57.14
PITT	RF	52.63	54.55	40.00
	SVM	50.00	50.00	34.48
	SAE	55.26	57.14	48.48
	Ours	57.69	92.30	75.00
LEUVEN	RF	53.45	62.50	27.03
	SVM	58.62	85.71	33.33
	SAE	65.52	84.62	52.38
	Ours	71.15	88.46	82.35
MAXMUN	RF	47.91	47.62	44.44
	SVM	56.25	57.89	51.16
	SAE	60.42	64.71	53.66
	Ours	61.90	76.19	66.66
KKI	RF	60.00	61.53	57.14
	SVM	66.66	64.71	68.75
	SAE	63.33	64.29	62.07
	Ours	66.66	75.00	70.00

## Data Availability

In this study, the data is from a public database, which is Autism Brain Imaging Data Exchange (ABIDE). The data is available at http://fcon_1000.projects.nitrc.org/indi/abide/ accessed on 24 October 2022.

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
