# Peer review of "A Similarity Measure-Based Approach Using RS-fMRI Data for Autism Spectrum Disorder Diagnosis"

_diagnostics, 2023, doi:10.3390/diagnostics13020218_

Round 1

Reviewer 1 Report

Dear editors of Diagnostics,

It has been a pleasure to review the manuscript by Zhang et al. Indeed, the subject of the manuscript is very interesting. Applying siamese networks as a strategy for classifying patients based on small samples of images is a timely and relevant subject. However, I have some general concerns regarding the lack of clarity and "organization" of the text. My specific points are:

1.- The abstract needs some improvement. It does not even mention that classifiers are applied to brain imaging data (resting state functional MRI data).

2.- Section 4.1. (Experimental setup) should be in Methods not in Results. Section 5.1. (Generalization performance on target sites) should be in Results not in Discussion. Section 5.2. (Training loss) should also be in Results.

3.- From my point of view, the most interesting and novel part of the manuscript is the Siamese network + Few shot approach. However, for someone with some knowledge of network classification but unfamiliar with this specific approach is very hard to get a clear idea from sections 3.3-3.4. How is the siamese network trained and later tested? What are positive / negative samples?  

4.- A new figure could also clarify the multisite protocol steps (including descriptions of baseline, meta-training sets, meta-test sets and the way the total loss is used to tune the parameters).  

5.- As inputs for each individual, average dynamic correlation matrices are computed. However, is this not equal, equivalent or very similar to directly calculating simple correlation matrices without time slide windows? If so, there is no need to make it that complicated.

4.- The proposed methodology is compared with other Machine Learning algorithms (SVM, RF and SAE), which, in turn are based on some hyperparameters and regularization steps. How where these hyperparametres and regularization values chosen? This should be explained.

6.- When the proposed method is compared with the other algorithms the YALE site is used as baseline. Since the YALE site was the one providing the highest accuracies in the metatest sets (Table 2), isn't this choice leading to positively biased results in the comparison?

7.- Finally, a set of new methods have been recently developed to homogenize brain images that were acquired in different scanners (see for instance, recent applications of the ComBat harmonization method in different image modalities). For this manuscript you could check the paper entitled "Statistical harmonization corrects site effects in functional connectivity measurements from multi-site fMRI data". it would be interesting to see how this harmonization impacts on the proposed methodology (and in the SVM, RF and SAE results).

Author Response

Responses to Reviewer

We thank the reviewers and editors for their very detailed and constructive comments. Our point-by-point response to those comments is provided below. We have also revised our paper accordingly and highlighted the changes in red for convenience of review. We also include related paper modifications after each response for easy reference.

  1. The abstract needs some improvement. It does not even mention that classifiers are applied to brain imaging data (resting state functional MRI data).

Response: Thank you for your suggestion. According to your suggestion, we have improved the abstract and clearly pointed out that the proposed is applied to RS-fMRI data in our revised paper (from line 6 to line 8 on Page 1), which is also copied below for your reference.

“Specifically, the few-shot learning strategy is used to measure the potential similarity in the RS-fMRI data distributions, and furthermore a similarity function for samples from multiple sites is trained to enhance the generalization.”

  1. Section 4.1. (Experimental setup) should be in Methods not in Results. Section 5.1. (Generalization performance on target sites) should be in Results not in Discussion. Section 5.2. (Training loss) should also be in Results.

Response: Thank you for your suggestion. According to your suggestion, we adjusted the structure of the paper.

  1. From my point of view, the most interesting and novel part of the manuscript is the Siamese network + Few shot approach. However, for someone with some knowledge of network classification but unfamiliar with this specific approach is very hard to get a clear idea from sections 3.3-3.4. How is the Siamese network trained and later tested? What are positive / negative samples?

Response: Thank you for your comment. According to your comment, we explained the training and testing process of the Siamese network in the revised manuscript (from line 116 to line 128 on Page 4-5). In addition, we have used NC/ASD as the category of samples in the paper, so we replace positive/negative samples with NC/ASD samples in the revised manuscript to avoid ambiguity (from line 140 to line 141 on Page 5), which is also copied below for your reference.

“As illustrated in Figure 2, the training Siamese network needs to input paired samples and optimize the model parameters by minimizing the loss function. In the settings of this section, the similarity is measured by distance (Eq.2), so when minimizing the loss function (Eq.3), the distance between paired samples from the same category should be small, while the distance between paired samples from different categories should be large.

In the test phase, the test dataset is divided into support set and query set. Suppose there are C categories in the data set and N samples in the support set of each category. Such few-shot learning classification task is called C-way N-shot task. Figure 3 shows the test flowchart of the trained Siamese network model. In our case, we have two categories (NC and ASD), so it is a 2-way N-shot task. Given a query sample Q from the query set, each support set has N support samples. The distances/similarities between the query sample Q and the samples in the two support sets are calculated to predict the label of sample Q.”

“It is worth noting that the number of NC samples in the baseline set may not be equal to the number of ASD samples.”

  1. A new figure could also clarify the multisite protocol steps (including descriptions of baseline, meta-training sets, meta-test sets and the way the total loss is used to tune the parameters).

Response: Thank you for your suggestion. According to your suggestion, we added a diagram to illustrate the training process in the revised manuscript (Figure 4 on Page 6), which is also copied below for your reference.

  1. As inputs for each individual, average dynamic correlation matrices are computed. However, is this not equal, equivalent or very similar to directly calculating simple correlation matrices without time slide windows? If so, there is no need to make it that complicated.

Response: Thank you for your comment. In the existing literature on functional connectivity, dynamic functional connectivity is widely recommended to measure the dynamic interaction between brain regions. Mean value is the basic method to extract features from dynamic functional connection, in addition, it also includes methods to extract features such as central moments. Although mean is the most basic method to extract features from dynamic functional connectivity, many literatures show that mean features are better than features without sliding windows. Therefore, we use the mean feature of dynamic functional connectivity in this paper.

  1. The proposed methodology is compared with other Machine Learning algorithms (SVM, RF and SAE), which, in turn are based on some hyperparameters and regularization steps. How where these hyperparametres and regularization values chosen? This should be explained.

Response: Thank you for your suggestion. According to your suggestion, the parameters of the comparison method are described in the revised manuscript, (from line 193 to line 196 on Page 7), which is also copied below for your reference.

“SVM and RF are implemented using Python based sklearn library, and default parameters are used. SAE consists of three full connection layers with tanh activation function, and uses two full connection layers for classification.”

  1. When the proposed method is compared with the other algorithms the YALE site is used as baseline. Since the YALE site was the one providing the highest accuracies in the metatest sets (Table 2), isn't this choice leading to positively biased results in the comparison?

Response: Thank you for your comment. In our approach, the baseline site runs through training and testing. This is because our goal is to find the most suitable site from the training site as the baseline, or we want to train a model that can achieve a good classification effect when comparing the similarity between the data from any site and the data from a specific site. In the test phase, the data in the baseline site is used as the support set. Therefore, in the training process, in addition to training model parameters, it is also necessary to find appropriate support sets for test stage. In this way, even if new data is added to the database in the future, it is still expected to achieve good performance on the new data. According to your comment, we explained in the revised manuscript (from line 134 to line 138 on Page 5), which is also copied below for your reference.

“Our goal is to train the model parameters so that the distance/similarity comparison between other site data and baseline site data can achieve satisfactory results. That is to say, in addition to training model parameters, a site should be found as the baseline site during training the model, which will be used as the support set in the test phase.”

  1. Finally, a set of new methods have been recently developed to homogenize brain images that were acquired in different scanners (see for instance, recent applications of the ComBat harmonization method in different image modalities). For this manuscript you could check the paper entitled "Statistical harmonization corrects site effects in functional connectivity measurements from multi-site fMRI data". it would be interesting to see how this harmonization impacts on the proposed methodology (and in the SVM, RF and SAE results).

Response: Thank you for your comment. The paper “Statistical harmonization corrects site effects in functional connectivity measurements from multi-site fMRI data” is an interesting study, we intend to combine ComBat harmonization method in our future research. And, we discuss this in the conclusion section in the revised manuscript (from line 238 to line 239 on Page 9), which is also copied below for your reference.

“In previous study, ComBat harmonization method was used to eliminate the impact of data heterogeneity among sites [43]. In future work, we consider combining small sample learning and ComBat to better eliminate the adverse impact of data heterogeneity between sites.”

References

[43] Yu, M.; Linn, K.A.; Cook, P.A.; Phillips, M.L.; McInnis, M.; Fava, M.; Trivedi, M.H.; Weissman, M.M.; Shinohara, R.T.; Sheline, Y.I. Statistical harmonization corrects site effects in functional connectivity measurements from multi-site fMRI data. Human Brain Mapping 2018, 39, 4213–4227.

Reviewer 2 Report

The manuscript presents a similarity based measure for ASD diagnosis based on fMRI-data. The manuscript is well written, presents a method that I find sound and based on the presented results, the proposed approach appear quite successful when compared to existing methods. In sum, I like this manuscript a lot and would like to congratulate the authors to work well done!

General comments:

ASD is commonly diagnosed based on behavioral queues using e.g. the ADOS score. As such, when first reading the abstract, I assumed that the data used for these methods would be behavioral data, not fMRI data. I would appreciate if the focus on brain imaging data was made more clear in the title, or at least int he abstract.

I also miss a proper background of the paper. The introduction does provide the most relevant references and create a short but clear framing of the paper. This is in itself not a bad thing, but I still think that a broader discussion of related methods (maybe drawn from analysis of other types of data) would be interesting. If the authors want to keep the introduction/background short, this broader framing could be added in the end in an extended conclusions section.

Specific points that needs clarification:

Sec. 3.1: ”The data were excluded when time points were missing from the original imaging data.” This is not very clear to me. How much data was excluded and what does it mean that the time points was missing?

Sec. 3.4: I find this section difficult to follow and at the same time important in order to understand how the overall classification (diagnosis) method work. Specifically, on line 100 ”The baseline set is used in both training and final performance test”. What does this mean? Did you use the same data for both training and test? And on 101 ”we use the data of a specific site as the baseline”. I do not understand, did you youse all data from one specific site as baseline, and if so, which? Or did you use some of the data from one site at a time and applied a standard round-robin approach to rotate? I, and maybe other readers, have no previous experience of the few-shot approach (C-way N-shot). I believe some more background on this would improve clarity.  Right now, I’m just assuming C are the 2-categories (NC and ASD) and N are the samples, but I do not know if N refers to the total number of samples in each site, number of positive/negative, whatever positive refers to in this context, etc. The details are hard to get to here.

Sec. 5: ”For consistency comparison, the rs-fMRI data from the five imaging sites (i.e., UCLA, UM, USM, NYU, YALE) were combined for training SVM, RF and SAE, the data from the target site is used to evaluate performance.” If I’m not missing something, this is all that is said on the comparison data. While I find the comparison itself fair and relevant, a reader needs much more to understand what this means. How was the data divided into training & test, and was parts of it used for validation in order to avoid overfitting? How does this division compare to your previous statement of the baseline data for your own method. That is, have your method seen more, less or exactly the same data than the other methods?

Fig 3. I was surprised about the labeling. Why not call loss0 for UCLA loss instead?

Minor comments:

Line 95: This makes when two subjects… makes what?

I believe the abbreviation NC is never defined (line 102).

Author Response

Responses to Reviewer

We thank the reviewers and editors for their very detailed and constructive comments. Our point-by-point response to those comments is provided below. We have also revised our paper accordingly and highlighted the changes in red for convenience of review. We also include related paper modifications after each response for easy reference.

General comments:

  1. ASD is commonly diagnosed based on behavioral queues using e.g., the ADOS score. As such, when first reading the abstract, I assumed that the data used for these methods would be behavioral data, not fMRI data. I would appreciate if the focus on brain imaging data was made more clear in the title, or at least in the abstract.

Response: Thank you for your suggestion. According to your suggestion, the title of the article was changed to “A similarity measure-based approach using RS-fMRI data for autism spectrum disorder diagnosis”. And, we have improved the abstract and clearly pointed out that the proposed is applied to RS-fMRI data in our revised paper (from line 6 to line 8 on Page 1), which is also copied below for your reference.

“Specifically, the few-shot learning strategy is used to measure the potential similarity in the RS-fMRI data distributions, and furthermore a similarity function for samples from multiple sites is trained to enhance the generalization.”

  1. I also miss a proper background of the paper. The introduction does provide the most relevant references and create a short but clear framing of the paper. This is in itself not a bad thing, but I still think that a broader discussion of related methods (maybe drawn from analysis of other types of data) would be interesting. If the authors want to keep the introduction/background short, this broader framing could be added in the end in an extended conclusions section.

Response: Thank you for your suggestion. According to your suggestion, we added the broader framing in the end in an extended conclusions section (in our revised paper from line 241 to line 245 on Page 9), which is also copied below for your reference.

“In addition, this work has yielded satisfactory results only using RS-fMRI data, but some recent studies have shown that it is potential to use multimodal data to diagnose neurological diseases, such as combining RS-fMRI data and structural MRI data for disease diagnosis. Using multimodal data to diagnose neurological diseases will be the focus of our future work.”

Specific points that needs clarification:

  1. Sec. 3.1: “The data were excluded when time points were missing from the original imaging data.” This is not very clear to me. How much data was excluded and what does it mean that the time points was missing?

Response: Thank you for your comment. In this study, we downloaded the preprocessed dataset from the C-PAC project. We downloaded 866 sample data from the 12 selected imaging sites, and we finally used 825 sample data, so we excluded 41 data. The “time points were missing” means that the scanning data at a certain time point is lost or damaged due to some reasons when the subject conducts fMRI scanning, such as excessive head movement and eye closure during scanning. To avoid ambiguity, we have revised the manuscript (in our revised paper from line 72 to line 74 on Page 2), which is also copied below for your reference.

“The data were excluded when the scanning data at some time points were missing from the original imaging data.”

  1. Sec. 3.4: I find this section difficult to follow and at the same time important in order to understand how the overall classification (diagnosis) method work.

Response: Thank you for your comments, and we response to those comments by point-by-point.

4.1 Specifically, on line 100 “The baseline set is used in both training and final performance test”. What does this mean? Did you use the same data for both training and test?

Response: We used the data from the baseline site in both the training and testing phase, one of the tasks of training models is to find a suitable baseline site. We revised the expression in the manuscript (in our revised paper from line 134 to line 138 on Page 5), which is also copied below for your reference.

“Our goal is to train the model parameters so that the distance/similarity comparison between other site data and baseline site data can achieve satisfactory results. That is to say, in addition to training model parameters, a site should be found as the baseline site during training the model, which will be used as the support set in the test phase.”

4.2 And on 101 “we use the data of a specific site as the baseline”. I do not understand, did you use all data from one specific site as baseline, and if so, which? Or did you use some of the data from one site at a time and applied a standard round-robin approach to rotate?

Response: We use a specific site as the baseline site. The baseline site is determined after model training. After the baseline site is determined, subsequent experiments use the same baseline. In section 4.2, we clarified the selected baseline site in the manuscript (from line 197 to line 198 on Page 7), which is also copied below for your reference.

“In the current experiment, the RS-fMRI data from YALE imaging site is selected as the baseline set for the proposed method”

4.3 and maybe other readers, have no previous experience of the few-shot approach (C-way N-shot). I believe some more background on this would improve clarity. 

Response: We explained C-way N-shot in the revised manuscript (from line 121 to line 128 on Page 4-5), which is also copied below for your reference.

“In the test phase, the test dataset is divided into support set and query set. Suppose there are C categories in the data set and N samples in the support set of each category. Such few-shot learning classification task is called C-way N-shot task. Figure 3 shows the test flowchart of the trained Siamese network model. In our case, we have two categories (NC and ASD), so it is a 2-way N-shot task. Given a query sample Q from the query set, each support set has N support samples. The distances/similarities between the query sample Q and the samples in the two support sets are calculated to predict the label of sample Q.”

4.3 Right now, I’m just assuming C are the 2-categories (NC and ASD) and N are the samples, but I do not know if N refers to the total number of samples in each site, number of positive/negative, whatever positive refers to in this context, etc. The details are hard to get to here.

Response: N is the number of samples in the support set. And, we have used NC/ASD as the category of samples in the paper, so we replace positive/negative samples with NC/ASD samples in the revised manuscript to avoid ambiguity (from line 140 to line 141 on Page 5), which is also copied below for your reference.

“It is worth noting that the number of NC samples in the baseline set may not be equal to the number of ASD samples.”

  1. Sec. 5: “For consistency comparison, the rs-fMRI data from the five imaging sites (i.e., UCLA, UM, USM, NYU, YALE) were combined for training SVM, RF and SAE, the data from the target site is used to evaluate performance.” If I’m not missing something, this is all that is said on the comparison data. While I find the comparison itself fair and relevant, a reader needs much more to understand what this means. How was the data divided into training & test, and was parts of it used for validation in order to avoid overfitting? How does this division compare to your previous statement of the baseline data for your own method. That is, have your method seen more, less or exactly the same data than the other methods?

Response: Thank you for your comment. We combine the data of the five sites used to train the proposed model into a data set and use it as a training set to train the comparison model. For comparison methods, training set and test set are independent. In the training process, 70% of the training samples are used to train the comparison methods parameters, and the remaining 30% are used to evaluate the performances of the models during training. In other words, our method will not see more data than the comparison method. We have revised the description of this part in the revised manuscript (from line 203 to line 206 on Page 7), which is also copied below for your reference.

“the data from the target site is used as a test set to evaluate performance. In the training process, 70% of the training samples are used to train the comparison methods parameters, and the remaining 30% are used to evaluate the performances of the models during training.”

  1. Fig 3. I was surprised about the labeling. Why not call loss0 for UCLA loss instead?

Response: Thank you for your comment. We originally randomly set the loss label, now the loss labels are set to UCLA-->loos0, UM-->loss1, USM-->loss2, NYU-->loss3 in the revised manuscript (Figure 5 on Page 7), which is also copied below for your reference.

Minor comments:

  1. Line 95: This makes when two subjects… makes what?

Response: Thank you for your comment. According to your comment, we found this to be a wrong statement. We have deleted this sentence from the revised manuscript.

  1. I believe the abbreviation NC is never defined (line 102).

Response: Thank you for your comment. According to your comment, we gave the full definition of the abbreviation NC when it first appeared (in our revised paper from line 28 to line 30 on Page 1), which is also copied below for your reference.

“Zhao et al. [13] presented a multi-view high-order functional connectivity network (FCN) based on RS-fMRI data for ASD vs. normal control (NC) classification.”

Reviewer 3 Report

Authors have proposed a similarity measure-based method for autism spectrum disorder diagnosis.

1-    In the abstract section, I would suggest that the author should provide the point and quantitative advantages of classification results.

2- In the result section, I would suggest that the author should provide a P-value for performance comparison between the proposed model and competing methods. 

3-  Some new references should be added to improve the literature review—for example, https://doi.org/10.1007/s00247-022-05510-8; https://doi.org/10.1155/2020/1357853.

Author Response

Responses to Reviewer

We thank the reviewers and editors for their very detailed and constructive comments. Our point-by-point response to those comments is provided below. We have also revised our paper accordingly and highlighted the changes in red for convenience of review. We also include related paper modifications after each response for easy reference.

  1. In the abstract section, I would suggest that the author should provide the point and quantitative advantages of classification results.

Response: Thank you for your suggestion. According to your suggestion, we added the point and quantitative advantages of classification results in the abstract section in our revised paper (from line 10 to line 14 on Page 1), which is also copied below for your reference.

“The experimental results show that the experimental indicators of the proposed method are better than those of the comparison methods in varying degrees, such as the accuracy on TRINITY site is more than 5% higher than that of the comparison method, which clearly proves that the presented approach achieves better generalization performance than the compared methods.”

  1. In the result section, I would suggest that the author should provide a P-value for performance comparison between the proposed model and competing methods. 

Response: Thank you for your suggestion. As far as we know, the P-value is obtained through statistical methods (e.g., t-test) and is the statistical result of two groups of data. In our method, the classification result is a single number, and we worry that it may not be possible to perform t-test on two different classifier results. We would appreciate your further suggestions about P-value statistics between different classifiers.

  1. Some new references should be added to improve the literature review—for example, https://doi.org/10.1007/s00247-022-05510-8; https://doi.org/10.1155/2020/1357853.

Response: Thank you for your suggestion. According to your suggestion, we added three references in our revised paper (from line 25 to line 27 on Page 1; from line 40 to line 45 on Page 2), which is also copied below for your reference.

In neuroimaging, resting-state functional magnetic resonance imaging (RS-fMRI) utilizes blood oxygen level dependent (BOLD) signals to explore biomarkers of nervous system diseases [8-11].(from line 25 to line 27 on Page 1)

“For instance, Brown et al. [20] proposed the element-wise layer for DNNs to predict ASD, without considering the heterogeneity of data from different sites. The other type aims to avoid the adverse effect of data heterogeneity on the results [21-23]. For example, Niu et al. [21] proposed a multi-channel deep attention neural network to capture the correlation in multi-site data by integrating multi-layer neural networks, attention mechanisms and feature fusion.” (from line 40 to line 45 on Page 2)

Added References:

[11] Ali, R.; Li, H.; Dillman, J.R.; Altaye, M.; Wang, H.; Parikh, N.A.; He, L. A self-training deep neural network for early prediction of cognitive deficits in very preterm infants using brain functional connectome data. Pediatric radiology 2022, 52, 2227–2240.

[20] Brown, C.J.; Kawahara, J.; Hamarneh, G. Connectome priors in deep neural networks to predict autism. In Proceedings of the 2018 IEEE 15th international symposium on biomedical imaging (ISBI 2018). IEEE, 2018, pp. 110–113.

[21] Niu, K.; Guo, J.; Pan, Y.; Gao, X.; Peng, X.; Li, N.; Li, H. Multichannel deep attention neural networks for the classification of autism spectrum disorder using neuroimaging and personal characteristic data. Complexity 2020, 2020.